# The Prediction of Transmission Towers' Foundation Ground Subsidence in the Salt Lake Area Based on Multi-Temporal Interferometric Synthetic Aperture Radar and Deep Learning

**Bijing Jin** [1,†], **Taorui Zeng** [2,†] , **Taohui Yang** [3], **Lei Gui** [1,*] , **Kunlong Yin** [1], **Baorui Guo** [1], **Binbin Zhao** [1,4] **and Qiuyang Li** [3]

1    Faculty of Engineering, China University of Geosciences, Wuhan 430074, China; begin@cug.edu.cn (B.J.); yinkl@cug.edu.cn (K.Y.); guobr@cug.edu.cn (B.G.); zhaobinbin@epri.sgcc.com.cn (B.Z.)
2    Institute of Geological Survey, China University of Geosciences, Wuhan 430074, China; zengtaorui@cug.edu.cn
3    State Grid Qinghai Electric Power Research Institute, Qinghai 810001, China; ythui0518@qh.sgcc.com.cn (T.Y.); lqyang2535@qh.sgcc.com.cn (Q.L.)
4    Research Institute of Transmission and Transformation Projects, China Electric Power Research Institute, State Grid Corporation of China, Beijing 100192, China
*    Correspondence: lei.gui@cug.edu.cn
†    These authors contributed equally to this work.

**Abstract:** Displacement prediction of transmission towers is essential for the early warning of transmission network deformation. However, there is still a lack of prediction on the ground subsidence of the tower foundation. In this study, we first used the multi-temporal interferometric synthetic aperture radar (MT-InSAR) approach to acquire time series deformation for the transmission lines in the Salt Lake area. Based on the K-shape clustering method and field investigation results, towers #95 and #151 with representative foundation deformation characteristics were selected for displacement prediction. Combined with field investigations and the characteristics of saline soil in the Salt Lake area, the trigger factors of transmission tower deformation were analyzed. Then, the displacement and trigger factors of the transmission tower were decomposed by variational mode decomposition (VMD), which could closely connect the characteristics of the foundation saline soil with the influence of the trigger factors. To analyze the contribution of each trigger factor, the maximum information coefficient (MIC) was quantified, and the best choice was made. Finally, the hyperparameters of the long short-term memory (LSTM) neural networks were optimized using a convolutional neural network (CNN) and the grey wolf optimizer (GWO). The findings reveal that the refined deep learning models outperform the initial model in generalization potential and prediction precision, with the CNN–LSTM model demonstrating the highest accuracy in predicting the total displacement of tower #151 (RMSE and $R^2$ for the validation set are 0.485 and 0.972, respectively). Given the scant research on the multifactorial influence on the ground subsidence displacement of transmission towers, this study's methodology offers a novel perspective for monitoring and early warning of ground subsidence disasters in transmission networks.

**Keywords:** transmission tower; ground subsidence; Salt Lake; displacement prediction; MT-InSAR; deep learning

## 1. Introduction

Transmission lines are critical facilities of the world power grid system, especially high-voltage transmission towers, which are essential for protecting the lives of inhabitants and preserving the stability of the economy [1,2]. In addition, owing to increasing electricity demand, the Qinghai–Tibet Plateau in Western China is regarded as an essential clean power energy base [3]. However, owing to fragile geological conditions and climate change,

the foundation of the 750 kV transmission towers in the Salt Lake area of the Tibet Plateau is particularly vulnerable to the impact of ground subsidence disasters. In recent years, severe deformation events have occurred in some transmission towers, seriously endangering the safety and effective operation of the transmission grid system in this region. Despite this, research on the ground subsidence of transmission tower deformation through medium- and long-term monitoring suggests that early warning research in the Salt Lake area is still limited. In the past, grid inspectors often conducted regular on-site surveys of each tower along a transmission line to monitor its operation. This approach was not only economically costly but also lacked insight into the overall deformation distribution of the line. Traditional monitoring and management practices were unable to effectively provide early warnings of tower deformations caused by ground subsidence.

Most ground subsidence is slow in development [4]. However, it is impractical to deploy monitoring equipment and conduct long-term field investigations. The multi-temporal InSAR (MT-InSAR) method, such as small baseline subset (SBAS) [5], permanent scatterer InSAR (PS-InSAR) [6], etc., have been widely used to measure ground deformation [7–9]. This method provides a low-cost and high-precision deformation monitoring method, especially in no man's land areas, where implementing only conventional geotechnical monitoring methods can be difficult [10–12]. Yan et al. [13] studied the use of the spaceborne SAR method to monitor power grid security in small-scale areas. Using the SBAS-InSAR method, Luo et al. [14] monitored the ground subsidence of the Yongshan County transmission line and obtained the ground deformation distribution. The InSAR early timing warning system is suitable for transmission line towers in the Salt Lake region of the Qinghai–Tibet Plateau. This is an entirely new attempt, limited to many transmission towers in the study area, resulting in differentiated time series and deformation characteristics.

In the early warning of transmission tower deformation, the tower displacement should be monitored along with the different trigger factors [15–18]. The change in the saline soil in the study area seriously influences the damage and deformation of transmission towers [19]. Water and temperature are considered the main factors affecting saline soil [20–22]. In saline soil, an increase in chloride ion content causes soil heaving when the temperature is below 17.9 °C [23]. Additionally, lengthening the freezing and thawing cycles would alter the soil strength of the saline soil [24]. Dry, saline soils have a high carrying capacity, but when water enters, it causes collapse and deformation [25]. Saline soil deforms due to underflow erosion when the water flow removes the salt and certain soil particles [26]. Additionally, as rainfall intensity increases, saline soil deforms more severely [26,27]. Upon analyzing the trigger factors of ground subsidence for transmission towers in the study area, a pivotal aspect to investigate is the methodology for quantifying the impact of various trigger factors. In previous studies, the maximum information coefficient (MIC) has been considered a reliable measure for determining the degree of association between two variables, with a value range of 0–1 [28,29]. This method is capable of identifying both linear and nonlinear functional relationships among variables (such as indices, periods, etc.) [30]. Consequently, in this paper, MIC values are employed to compare and sift through the principal trigger factors affecting the ground subsidence of the towers in the designated study area.

The mid-long-term warning of transmission towers requires the analysis and prediction of time series displacement. Some researchers have tried to divide time series displacements into short-term deformation trends and long-term deformation fluctuations [28,31]. The long-term deformation trend reflects the creep behavior of the soil under gravity or a continuous external force. Short-term deformation fluctuations are caused by sudden changes in external trigger factors [29,32]. At present, few studies have focused on time series research of transmission towers, especially in the unique geological environment of the Salt Lake area. This study uses a robust time series decomposition algorithm called variational mode decomposition (VMD). It has found many applications in finance, medicine, and energy [33]. Some studies have emphasized the application of

feature engineering methods to time series prediction. Factors with low contributions reduce the generalizability of the model [29,30]. Therefore, the maximum information coefficient algorithm [30] was used to consider the correlation between the trigger factors and time series.

A number of studies have suggested a variety of forecast methods, such as quasi-3D seepage models, 1D numerical methods, statistical methods, and machine learning, to warn against ground subsidence [4,34–36]. Since the rapid development of machine learning for data analysis, deep learning has become increasingly popular in prediction analysis [37]. Kim et al. [38] employed a CNN–LSTM neural network model to estimate the amount of energy consumed at home, and the results may have met the prediction objectives. Based on CNN, LSTM, and CNN–LSTM algorithms, Yan et al. [39] compared the air quality index prediction of multiple hours and multiple sites in Beijing and found that the CNN–LSTM model had an optimal effect. Mahmoodzadeh et al. [40] used the GWO–LSTM model to predict the basic physical and mechanical parameters of rocks and achieved exciting prediction results and generalization ability. However, research results using deep learning to predict the time series of ground subsidence on transmission towers have not been published.

The most critical research initiatives involve deformation monitoring and early warning of transmission towers in the Salt Lake region of the Qinghai–Tibet Plateau. This work considers the MT-InSAR method and deep learning algorithms to conduct innovative research on the ground subsidence of transmission line towers in the Salt Lake area. Time series displacement findings and trigger factors were decomposed using the VMD approach. Based on the MIC values, the main trigger factors for tower ground subsidence were selected. Finally, the CNN and GWO algorithms were used to optimize the LSTM model. Furthermore, the prediction results of the ground subsidence of the towers under different optimization models were compared and analyzed. The optimal prediction strategy can be used for the medium- and long-term early warning of transmission towers.

## 2. Geographical and Geological Setting

The transmission line is located in the Eastern Qaidam Basin of the Qinghai–Tibet Plateau and passes through Qarhan Salt Lake from south to north (Figure 1a). The line's length is about 170 km, and the elevation is between 2684 and 3488 m. In the natural division, the transmission line belongs to the cold zone of the Qinghai–Tibet Plateau and part of the Central Asian desert [20]. Affected by the Qilian Mountains and the Kunlun Mountains, it is difficult to enter warm and humid areas, which makes the transmission line dry and causes large evaporation [9]. The average annual precipitation is only 25 mm, whereas evaporation is as high as 3066 mm [41]. However, due to climate change, on the Tibetan Plateau, extreme rainfall events still occur [42]. Affected by the inland desert climate, the temperature difference between day and night in this area is large, and the maximum daily temperature difference reaches 30.6 °C. Owing to the seasonal melting of alpine snow and ice, there is a larger amount of water in summer than in winter, which also leads to complex surface gullies [9]. The groundwater depth of the transmission tower ranges from 3.5 to 4.2 m. Unfortunately, the transmission tower in the study area has not been installed with groundwater monitoring equipment, and effective groundwater data have not been obtained.

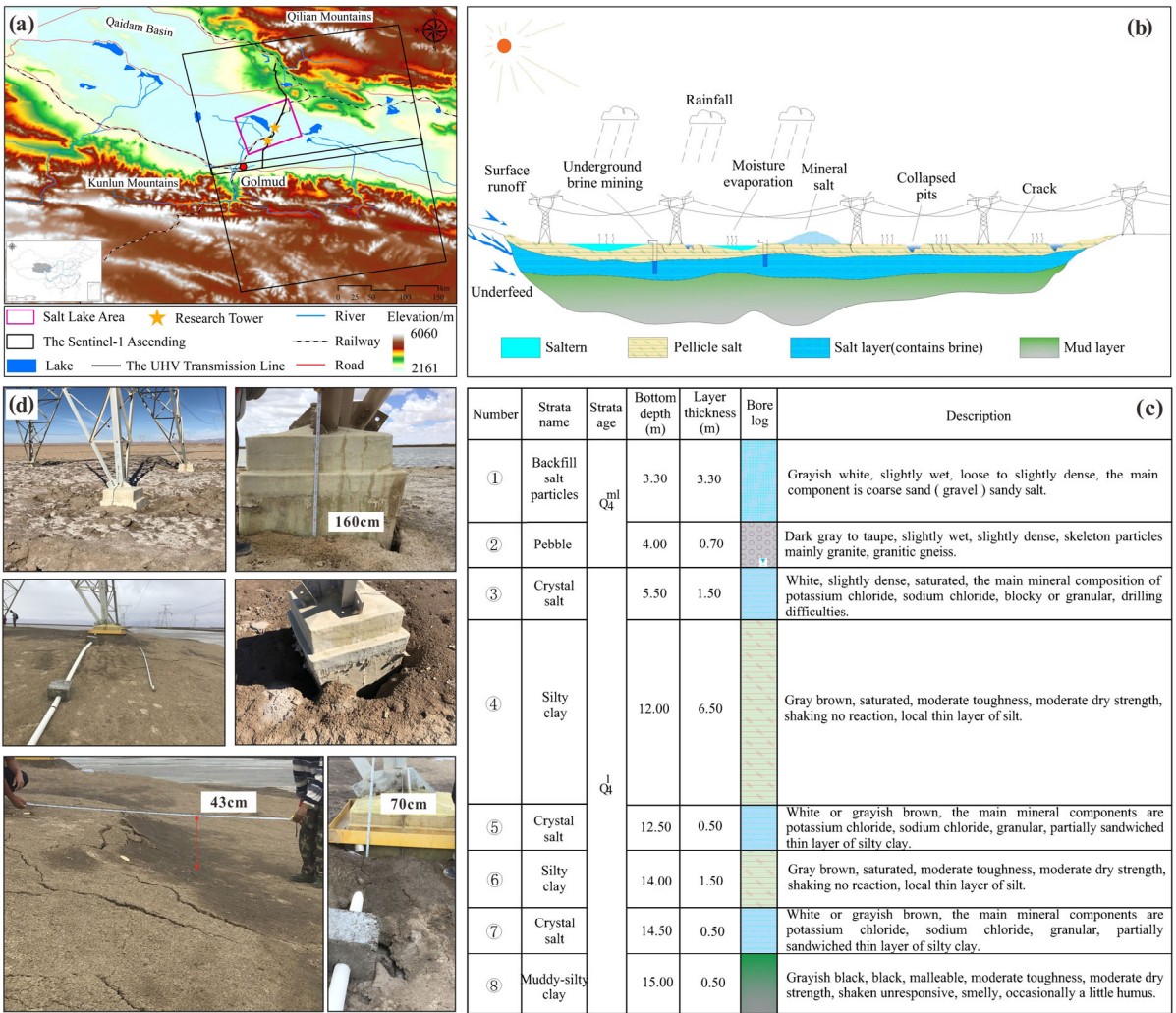

**Figure 1.** (**a**) Location of the study area. (**b**) The Salt Lake mineral salt production and tower deformation diagram. (**c**) The tower foundation drilling histogram in the Salt Lake area. (**d**) The field investigation results of the research tower include transmission tower collapse, rain erosion, deformation damage to drainage equipment, and so on.

Saline soils are widely distributed along the line and are often observed on the ground surface (Figure 1d). Different from the three-phase structure of normal soil, saline soil has a four-phase structure due to its 'soluble salt phase' [43,44]. When the water content is meager, supersaturated salts exist in the form of crystals between the soil particles and play a certain skeletal role [45]. However, when external conditions such as rainfall occur, the soluble salt in the soil is completely dissolved, the soil structure is damaged, the strength is reduced, and disaster-causing problems such as swelling, slurry, sinking, and corrosion are very likely to occur [20,46]. The ground survey results of transmission lines in the Salt Lake area show that dissolution fissures on the surface are developed, and the depth is between 0.1 and 0.2 m. According to the pre-drilling data, the foundation materials for the transmission line tower in the Salt Lake area are pebbles, crystalline salt, silty clay, and muddy silty clay from top to bottom (Figure 1b,c). Through indoor soluble salt detection, it was found that the crystalline salt of tower foundations in the Salt Lake area is mainly chlorine salt, which easily collapses after encountering water [21,43,47]. With the warming of the Qinghai–Tibet Plateau intensifying, alpine snowmelt may cause rising groundwater levels and frequent extreme rainfall events [48], and the safe operation of transmission line towers will be significantly threatened.

The transmission line was built in 2011 and started operating in 2013. According to the transmission line tower early patrol data, there was no evident deformation in the early stage of the operation. However, due to the significant increase in precipitation, ground subsidence around the transmission towers continued to develop. The most severe deformation occurred between 2017 and 2018. To date, more than 200 towers have been deformed and damaged owing to different degrees of land subsidence, and the number is still increasing. Field survey results revealed that different transmission towers in the Salt Lake region have varying degrees of ground collapse and cracking. There is still continuous ground subsidence deformation around the towers (Figure 1d).

## 3. Methodology

The framework for predicting the ground subsidence of the transmission tower is shown in Figure 2.

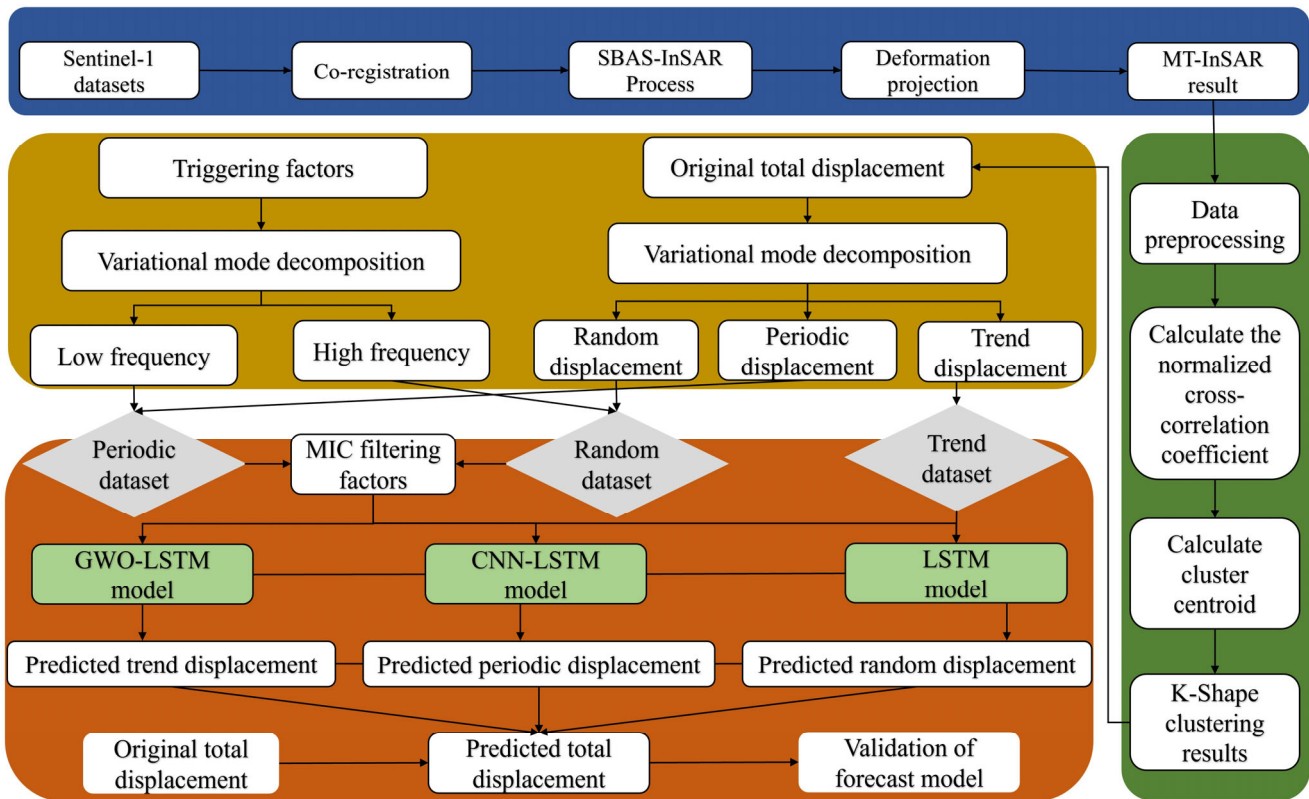

**Figure 2.** Flowchart of the forecast model in ground subsidence.

(a) The Sentinel-1A SAR datasets from 2017 to 2021 were gathered in the first stage, and SBAS-InSAR technology was used to obtain the same ground subsidence time series findings for transmission towers as the MT-InSAR approach.

(b) Based on the MT-InSAR results, K-shape clustering was used to analyze the time series characteristics of the ground subsidence of the transmission towers.

(c) To research the ground subsidence deformation trend of the representative towers, the time series results were decomposed by VMD to obtain the periodic, trend, and random displacements. The ground subsidence trigger factors of the transmission tower were decomposed using the VMD method. Similar to the high-frequency random displacement, the low-frequency sequences match the periodic displacement.

(d) Correlation analysis of the decomposed trigger factors was performed using the MIC algorithm to obtain the maximum correlation factor of transmission tower ground subsidence. Based on the above trigger factors and displacement decomposition

results, the CNN–LSTM, GWO–LSTM, and LSTM models were used for displacement prediction. Finally, the predicted results were validated using $R^2$ and the RMSE.

*3.1. SBAS-InSAR Method*

In 2002, a novel InSAR time series analysis technique (SBAS-InSAR) was presented by Berardino et al. [5] for tracking the temporal development of surface deformations. In this study, the SBAS-InSAR technique was used in MT-InSAR for ground settlement deformation analysis. The main technical principle of the SBAS-InSAR algorithm is as follows [5,49]: First, it is assumed that $N$ SAR images of the same region were obtained at times $t_1$, $t_2$ ..., $t_n$, and one image is selected randomly as the main image for registration. According to the interference combination condition, $M$ interference fringe patterns are formed under the condition of a short baseline distance, and $M$ satisfies:

$$\frac{N}{2} \le M \le \frac{N(N-1)}{2} \tag{1}$$

By performing three-dimensional space-time phase unwrapping on $M$ interference fringe patterns, it is possible to determine the deformation ($d_{LOS}$) of various SAR acquisition timings. By further projecting the ($d_{LOS}$) findings, the ground deformation results of the transmission line tower ($d_V$) can be obtained [50].

$$d_v = \frac{d_{LOS}}{\cos\theta} \tag{2}$$

For the SBAS-InSAR calculation, 67 × 2 (2 images for each date) Sentinel-1A images were obtained. The detailed data parameters of the images are listed in Table 1.

**Table 1.** The SAR dataset information about the transmission lines in the Salt Lake area.

| Satellite | Time-Span | Image Number | Off-Nadir Angle (°) | Azimuth Angle (°) | Resolution (Rg × Azi m) |
|---|---|---|---|---|---|
| Sentinel-1A | 2017/04/28–2021/12/8 | 67 × 2 | 33.7 | −10.4 (Ascending) | 2.33 × 13.97 |

*3.2. K-Shape Clustering of Time Series*

As a time series clustering method, the K-shape algorithm is widely used to classify time series data because of its efficiency and accuracy [51]. The K-shape algorithm uses a different distance measure and cluster center calculation than the K-means algorithm, which makes it effective in clustering time series. A distance metric based on the morphological similarity of the time series was used to extract the most representative morphological features from the time series. To ensure the accurate extraction of the form of the time series, the K-shape views the centroid computation as an optimization issue. The objective function is:

$$\vec{u}_k^* = \underset{\mu_j}{\mathrm{argmax}} \sum_{x \in C_j} (NCC_c(\vec{x},\vec{\mu}_j))^2 \tag{3}$$

where $C_j$ is the $j_{th}$ cluster and $j$ is the initial centroid of the $j_{th}$ cluster.

*3.3. Deep Learning and Optimization Model*

3.3.1. LSTM Model

To address the problems of gradient expansion or disappearance in the recurrent neural network (RNN), Hochreiter and Schmidhuber et al. [52] proposed an extended short-term memory network. A new deep learning neural network called long short-term memory networks (LSTM) extends the RNN input, output, and forget gates (Figure 3). Long-term temporal correlations can be captured by LSTM because of its selective filtering technique [31,52,53].

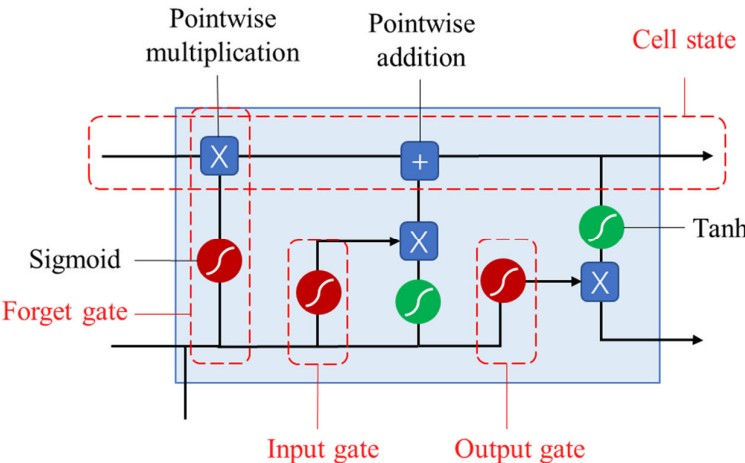

**Figure 3.** The structure of the LSTM network model.

### 3.3.2. CNN–LSTM Model

The convolutional, fully connected, and pooling layers make up the convolutional neural network (CNN) [39]. The formula for the convolutional layer feature map, which is typically used to calculate time series data, is as follows:

$$C = f(X \otimes W + B) \tag{4}$$

where $C$ is the feature map after the convolution kernel and $f$ denotes the rectified linear unit's (ReLU's) nonlinear activation function.

As a hybrid network, CNN–LSTM is widely used in time series prediction because it combines the merits of LSTM and CNN [39,54,55]. In this study, the convolutional and pooling layers of the CNN model were used to filter the main features from the input layer data and obtain a time-dependent sequence. The fully connected layer flattened the sequence before feeding it into the LSTM layer for time series displacement prediction. The dropout layer was then applied to prevent overfitting of the data, and the results of the predictions were produced. The structure of the CNN–LSTM model is shown in Figure 4.

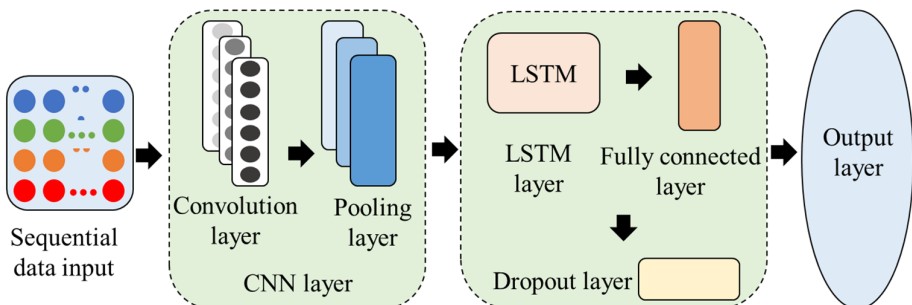

**Figure 4.** The structure flowchart of the CNN–LSTM model.

### 3.3.3. GWO–LSTM Model

Mirjalili [56] presented a novel meta-heuristic method called the grey wolf optimizer (GWO), which benefits from easy computation and a robust ability to search globally. The number of grey wolf generations and groups is defined by the GWO algorithm, which then determines the optimal parameters. To establish the fitness position, three head wolves ($\alpha$, $\beta$, and $\delta$ wolves) were employed. The remaining wolves compute the distance between themselves and the prey based on the location of the prey at the same time. Finally, the wolves gradually shorten the distance between the prey through continuous updating and evolution and hunt the prey [29,57].

The number of hidden layer neurons, learning rate, and number of iterations significantly influence the capacity of the LSTM model to fit data and make predictions. Consequently, the GWO technique was developed to optimize the hyperparameters of the LSTM model in the transmission tower ground subsidence displacement prediction. The structure of the model is illustrated in Figure 5.

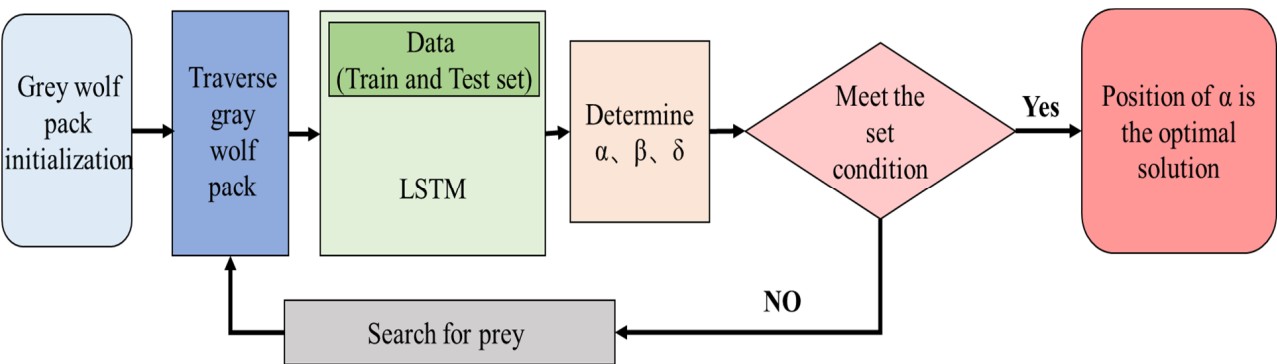

**Figure 5.** The structure flowchart of the GWO–LSTM model.

## 4. Results

### 4.1. Transmission Tower Ground Deformation Results

Using the MT-InSAR method, deformation maps from 2017 to 2021 in the transmission towers were created to show the spatial distribution of ground subsidence (Figure 6a). It can be seen from Figure 6a that the ground subsidence of the transmission towers is largely in the central Salt Lake region. The maximum ground subsidence value in the study area was –321 mm. Part of the tower was located in the center of the ground subsidence funnel (Figure 6b,c). By adding a 30 m buffer to the center of the tower, the values of the towers were obtained from the transmission line tower deformation [19], as shown in Figure 6d. There were two towers with the largest cumulative deformation: tower #95 (–89 mm) and #151 (−66 mm) (Figure 6d). A comparison of the UAV aerial photographs of the two towers shows that tower #95, in the operational phase, is located in the salt pond (the surface contains brine). Tower #151 is located in an abandoned salt pond (the surface is dry and presents a plowing landform). According to the results of the site investigation, the foundations of both tower 95# and tower 151# have a certain degree of settlement deformation, which has a greater impact on the stability of the towers. Therefore, accurate prediction of ground subsidence deformation of the transmission towers is of great significance to the operation and management of the subsequent transmission network. The interpreted surface deformation of some towers was positive, which may be due to the accumulation of wind and sand in the study area [58]. It has little or no effect on the stability of the tower.

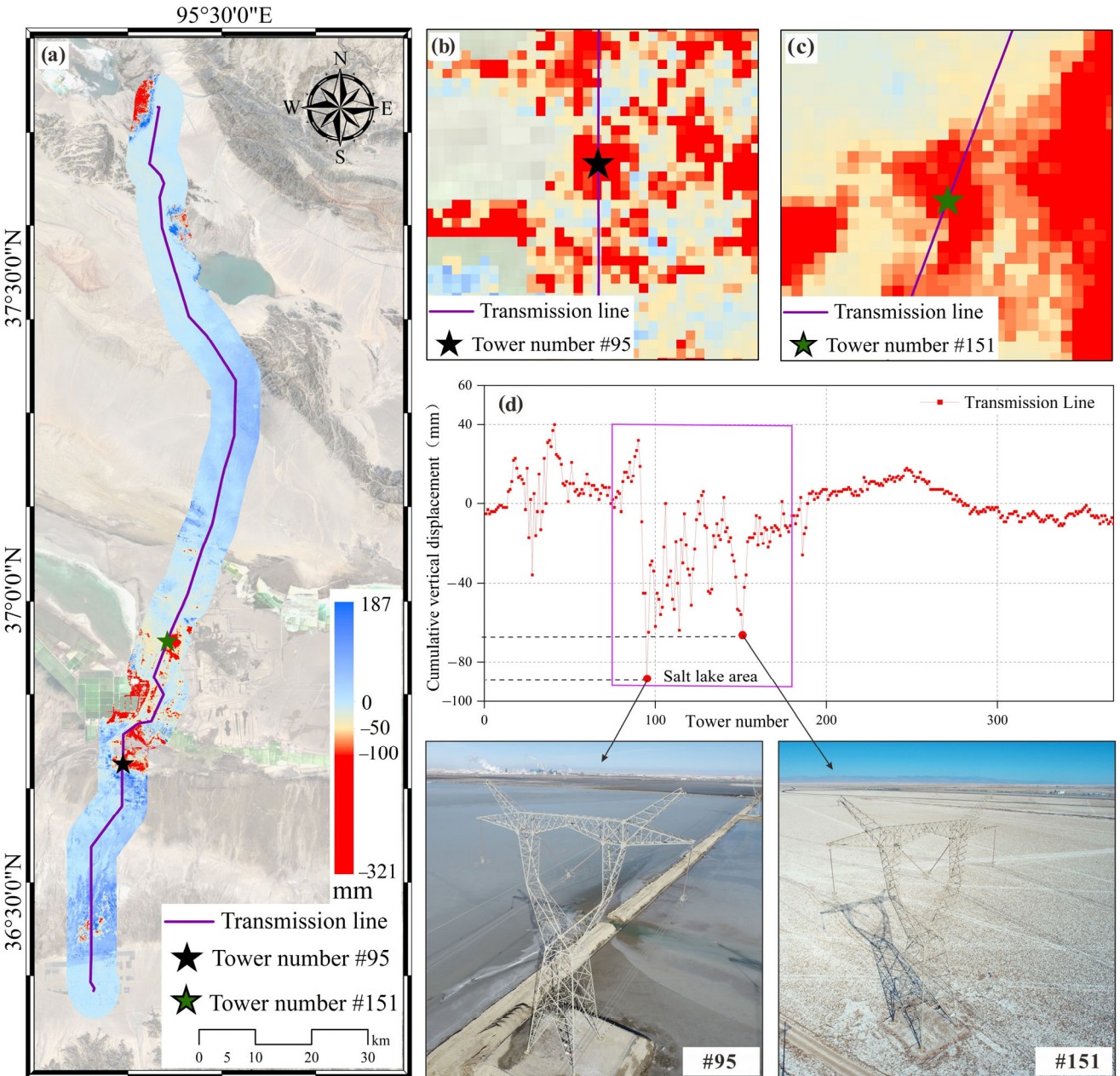

**Figure 6.** (**a**) The transmission line ground deformation remote sensing interpretation results. (**b**) The result of tower #95. (**c**) The result of tower #151. (**d**) The results of transmission towers and UAV aerial photos of towers #95 and #151.

*4.2. Ground Deformation Characteristic K-Shape Clustering Analysis Results*

The findings of the time series clustering based on K-shape clustering were obtained to further evaluate the time series deformation characteristics of each tower (Figure 7a–d). It could be seen that the tower time series deformation results were mainly divided into three types: overall upward type (Figure 7b), horizontal fluctuation type (Figure 7c), and overall downward type (Figure 7d). Through field investigation of the transmission line tower, it was found that the overall upward type is mainly located in the desert area, and its deformation is mainly caused by the accumulation of wind and sand (Figure 7e). The horizontal fluctuation type is mainly located in the wilderness area, and the horizontal fluctuation shown on the surface may have been caused by a slight disturbance of the surface (Figure 7f). The downward type is primarily found in the Salt Lake region, which is the main historical deformation point (Figure 7g).

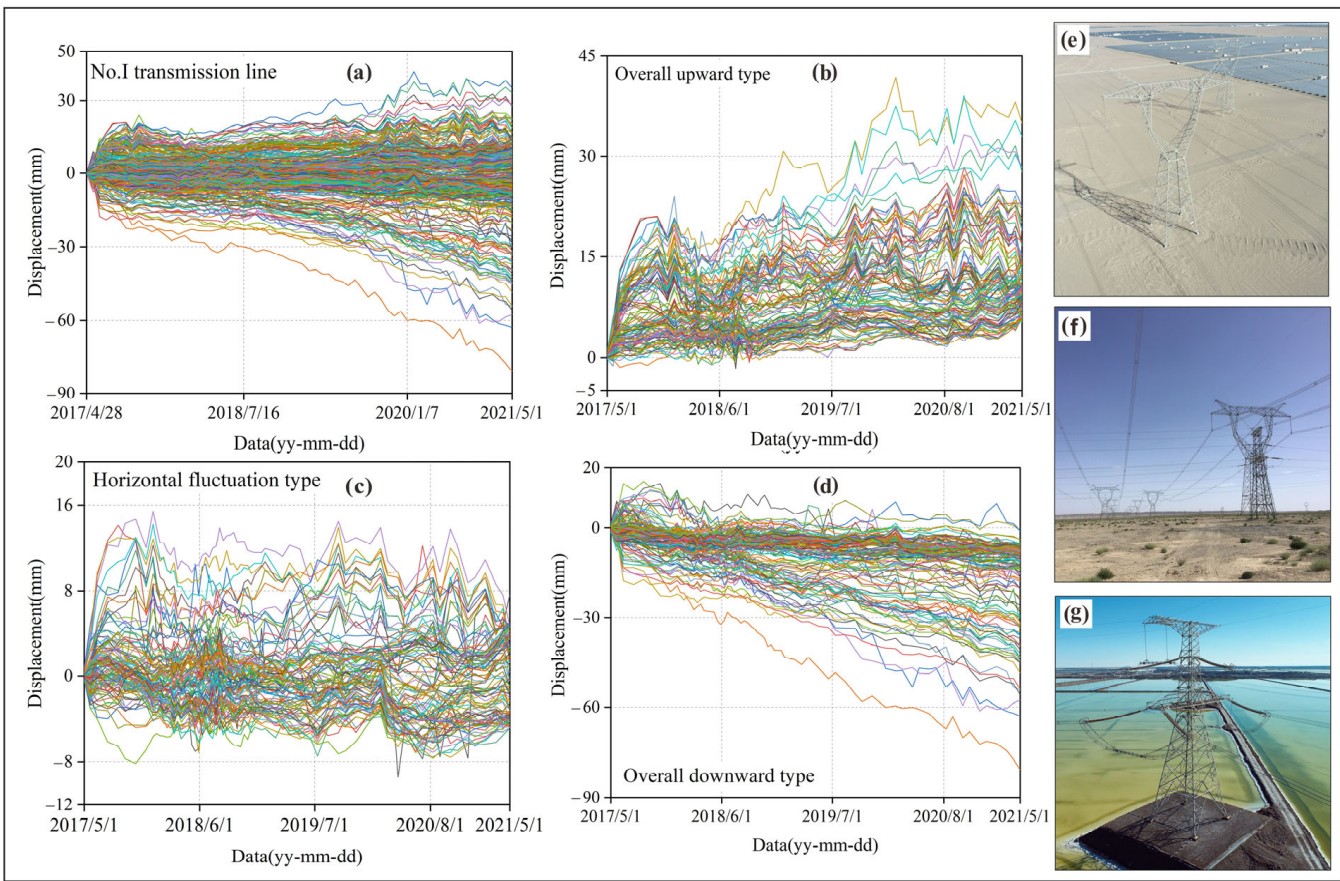

**Figure 7.** The results of the K-shape clustering, (**a**) transmission tower time series deformation results, (**b**) deformation results for the overall upward type of transmission tower time series (site environment as shown in (**e**), (**c**) deformation results for the horizontal fluctuation type of transmission tower time series (site environment as shown in (**f**), (**d**) deformation results for the overall downward type of transmission tower time series (site environment as shown in (**g**).

### 4.3. Decomposition of the Ground Displacement

MT-InSAR is a high-precision monitoring method that can be used as reliable primary data for ground displacement prediction in the absence of surface GPS monitoring equipment [59–61]. Based on the distribution of rainfall and temperature data obtained in the study area, 61 deformation values of towers #95 and #151 for the period from 28 April 2017 to 8 December 2020 were selected for the study. From Figure 8, it can be observed that both towers exhibit continuous downward displacement deformation during the interpretation period. This indicates that the towers will still exhibit a deformation trend in the future. Compared with #151, the time series results for #95 showed a slight fluctuation trend with increasing temperature and rainfall. This may be related to the evaporation and crystallization of the brine in the salt pond.

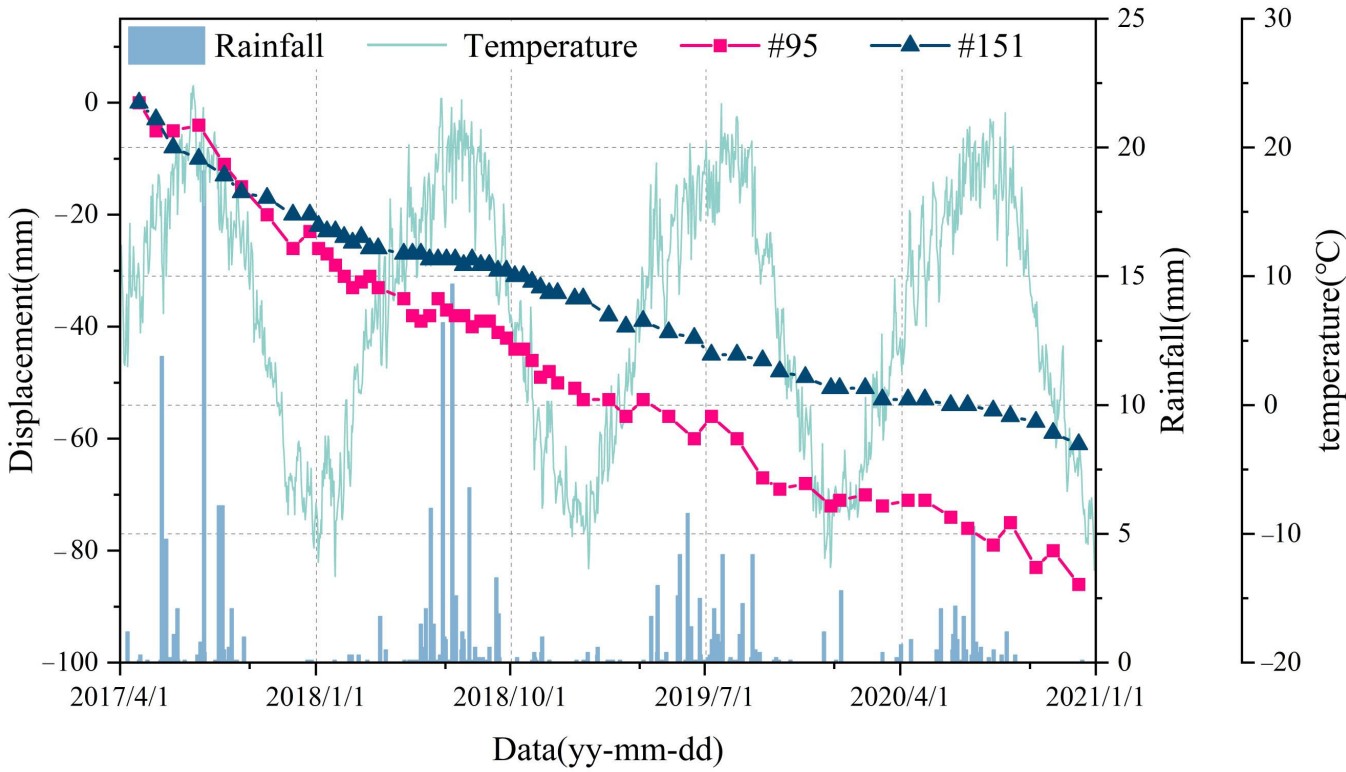

**Figure 8.** Monitoring data of rainfall, temperature, and the MT-InSAR results of towers #95 and #151.

In order to visualize the ground subsidence prediction model, an absolute valuation of the transmission tower ground subsidence displacement values was carried out. The thirty-seven groups from 28 April 2017 to 7 December 2018 were used to create the training samples. Subsequently, twelve groups from 31 December 2018 to 20 November 2019 were selected as test samples to evaluate the precision of the model and determine the optimal prediction model. To examine the viability of the implementation of the ideal prediction model, twelve groups' data from 7 January 2020 to December 2020 were employed as a verification set. In order to avoid leakage of data information, the validation set is not involved in the training of the model. The displacements of towers #95 and #151 were decomposed into trend, random, and periodic displacements using the VMD method (Figure 9). The results of the trend items were then compared. It could be found that the deformation characteristics of the two towers are the same. Both towers experienced a slight acceleration in 2018, which coincided with the actual deformation observed in 2018. From the decomposition results, #95 has more periodic deformation characteristics than #151, which may be related to its being in the salt pond. Regarding the deformation of the random term, both were minor.

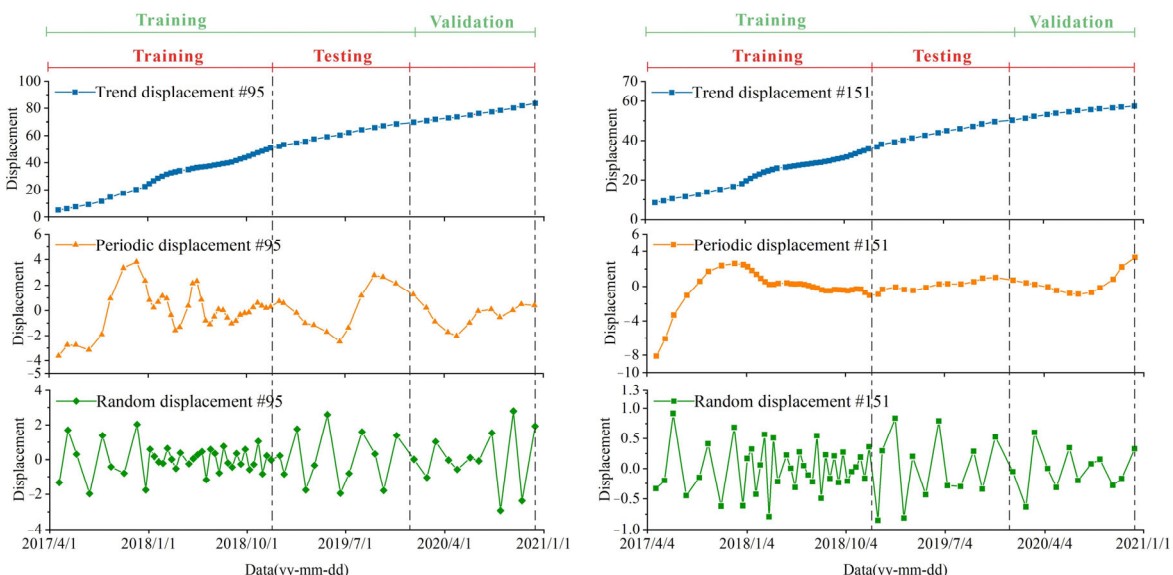

**Figure 9.** The results of the decomposition of the time-series displacements of towers #95 and #151.

### 4.4. Decomposition of Triggering Factors and Correlation Analysis

The temperature was used as an indirect intensity factor for surface evaporation and groundwater recharge in the study area. Combined with the deformation data time span, the temperature difference between Sentinel-1A images of two scenes was selected for research Figure 10 (F1). Rainfall is the primary trigger factor for transmission tower ground deformation. This study selected the cumulative rainfall for the day, the last two days, and the three days for analysis (F2, F3, and F4). The displacement changes during the last image (F5) and the final two photographs were chosen as the displacement factors (F6).

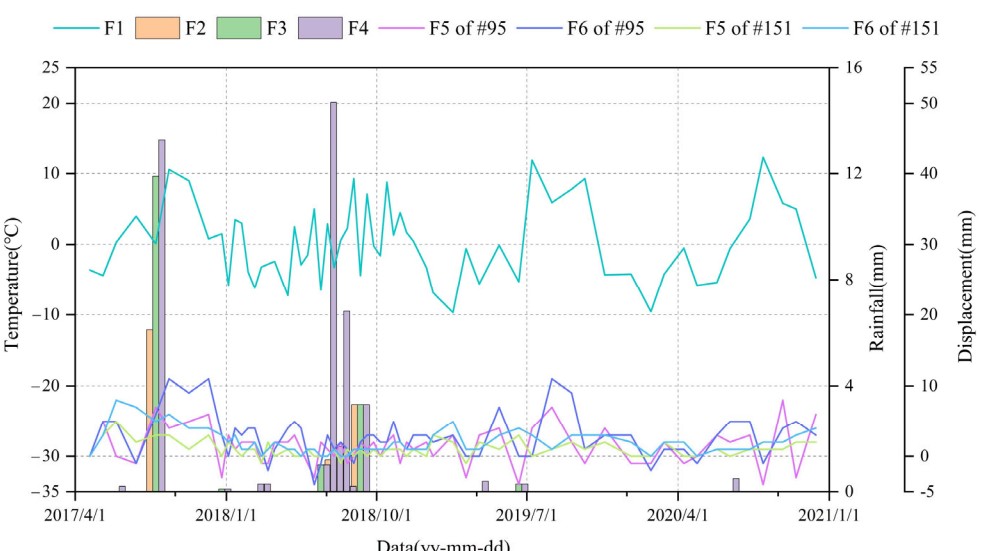

**Figure 10.** The candidate triggering factors of the ground subsidence displacement prediction. (F1: the change of the temperature during the last two Sentinel-1A images; F2: the cumulative of the rainfall for the day; F3: the cumulative of the rainfall for the last two days; F4: the cumulative of the rainfall for the three days; F5: the displacement changes during the last image; and F6: the displacement's change over the previous three Sentinel-1A images.)

The trigger factors were divided based on the VMD approach to produce high- and low-frequency sequences (Figure 11). Then, the high-frequency component was utilized as the

trigger factor for the random term displacement, whereas the low-frequency factor acquired by decomposition was used as the trigger factor for the periodic term displacement [28].

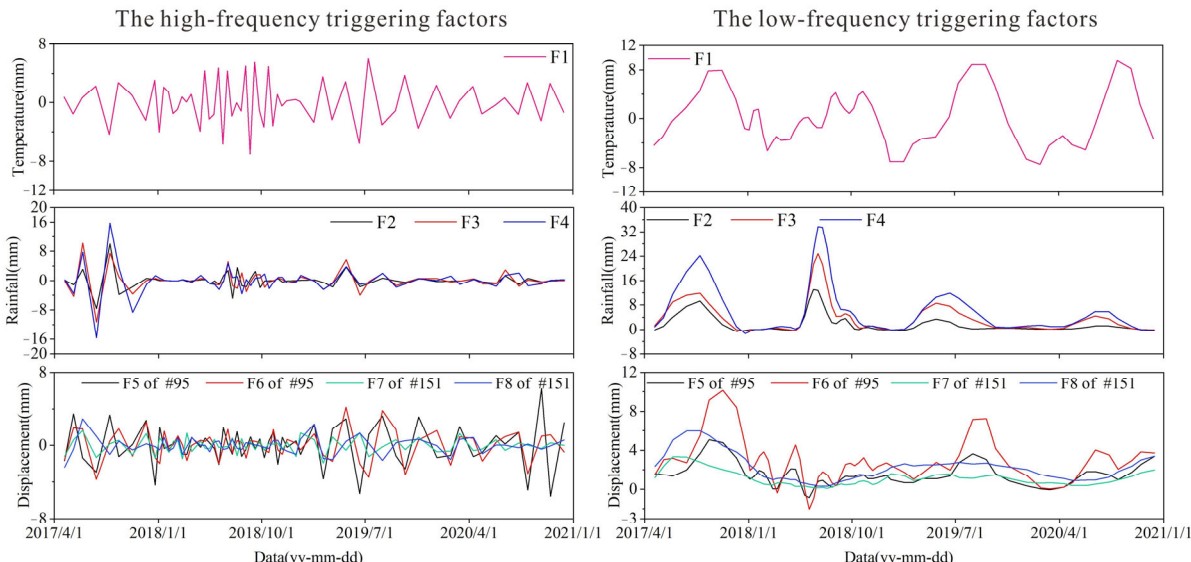

**Figure 11.** The candidate low-frequency and high-frequency triggering factors.

Six triggers were chosen as indicators for evaluating transmission tower displacement. There are different contributions to displacement prediction modeling for different trigger factors. The highest mutual information value for each trigger factor with transmission tower movement was determined using the MIC method (Figure 12). The greater the MIC value, the stronger the association between the two variables [29]. From Figure 12, it can be seen that #95 was more sensitive to temperature changes and rain than #151 (F1). This is also compatible with the brine evaporation and crystallization conditions that exist for #95 inside the salt pond. Based on the MIC values, F1, F2, and F5 were selected for random displacement prediction, and F1, F4, and F6 were selected for periodic displacement prediction.

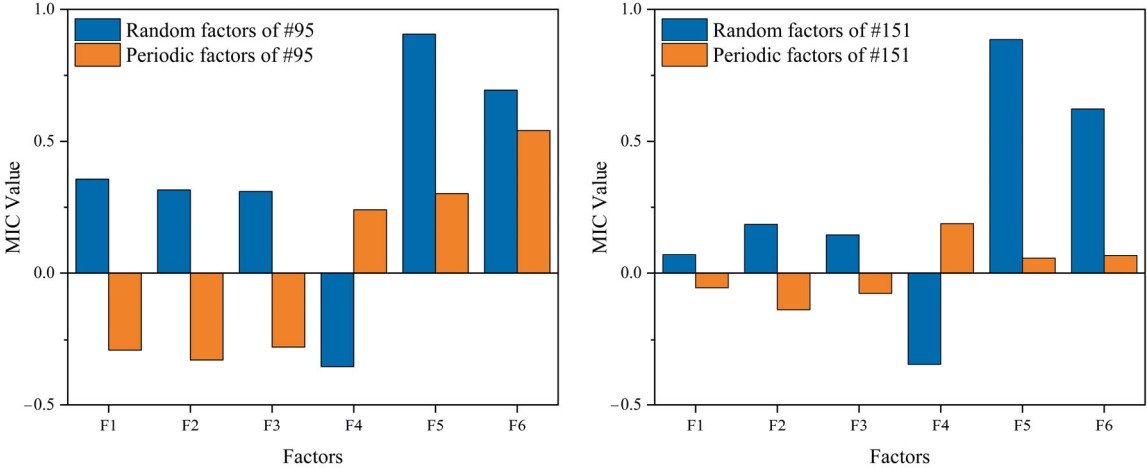

**Figure 12.** The MIC value of triggering factors.

### 4.5. Trend Displacement Prediction

The primary trend of the transmission tower ground subsidence was represented by the displacement trend. Therefore, this study used univariate GWO–LSTM, CNN–LSTM, and LSTM to predict the displacement trend. The ideal models were created utilizing the training and testing sets to prevent the leakage of data from the validation set. The

validation set was then predicted using the best models, and the training and test sets were merged to create a new training set for the validation set. The predicted results are shown in Figure 13, and the RMSE and $R^2$ of the prediction model for the trend displacement are listed in Table 2. Overall, the GWO and CNN-optimized LSTM performed better in both the training and validation sets, which can be used for the long-term prediction of the transmission tower ground subsidence trend term.

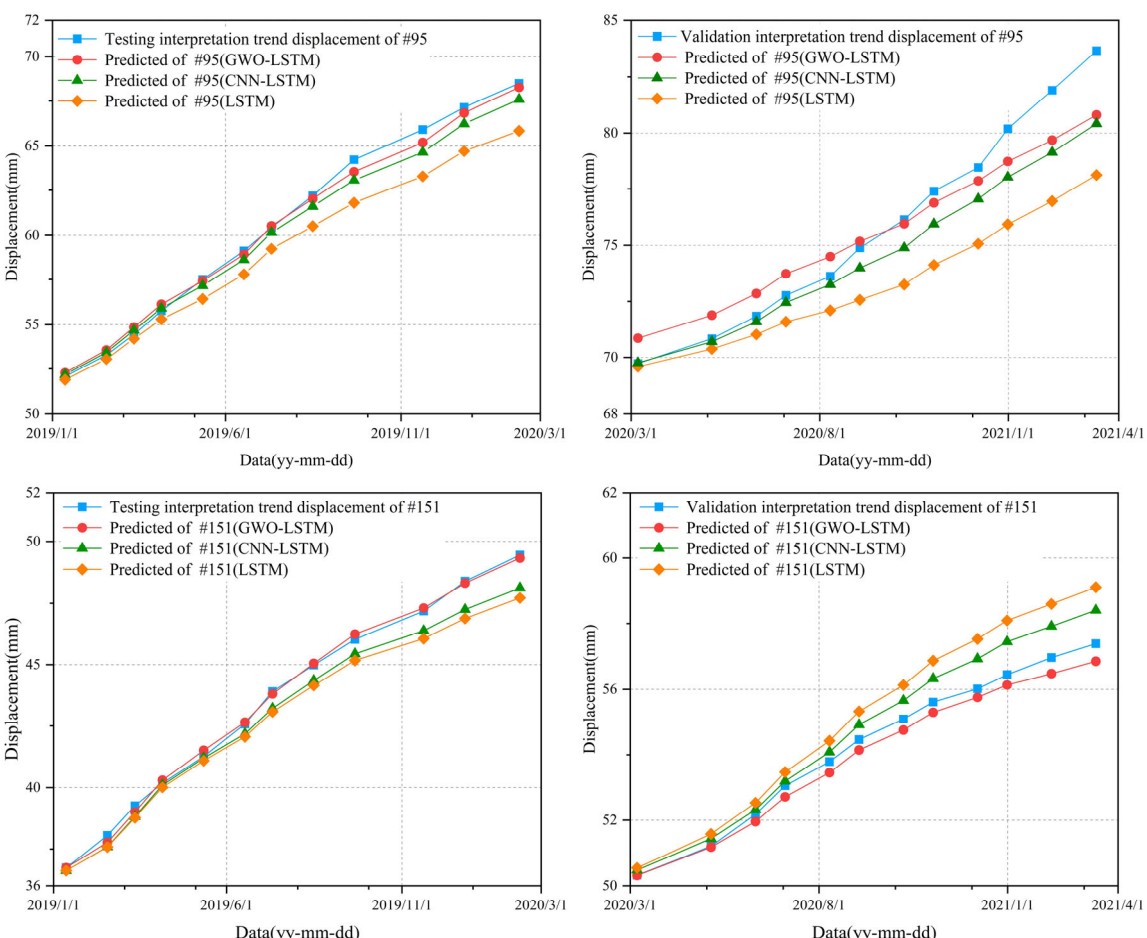

**Figure 13.** The trend displacement prediction results.

**Table 2.** The trend displacement prediction accuracy of towers #95 and #151.

| Tower Number | Model | Testing RMSE | Testing $R^2$ | Validation RMSE | Validation $R^2$ |
|---|---|---|---|---|---|
| #95 | GWO–LSTM | 0.368 | 0.995 | 1.324 | 0.904 |
| | CNN–LSTM | 0.674 | 0.984 | 1.570 | 0.862 |
| | LSTM | 1.664 | 0.904 | 3.079 | 0.482 |
| #151 | GWO–LSTM | 0.163 | 0.998 | 0.326 | 0.978 |
| | CNN–LSTM | 0.679 | 0.971 | 0.647 | 0.914 |
| | LSTM | 1.664 | 0.904 | 1.572 | 0.865 |

### 4.6. Periodic and Random Displacement Prediction

The periodic term displacement is caused by periodic action [62]. The effects of different trigger factors on the deformation of the transmission tower are different. According to the calculated MIC values, it was found that the periodic trigger factor of #95 was more significant than that of #151. The periodic displacement of the transmission tower was predicted using multiple factors based on the trigger factors F1, F2, and F5, which were

chosen in the last comparison. The prediction results are shown in Figure 14, and the RMSE and $R^2$ values of the prediction model for the displacement trend are listed in Table 3.

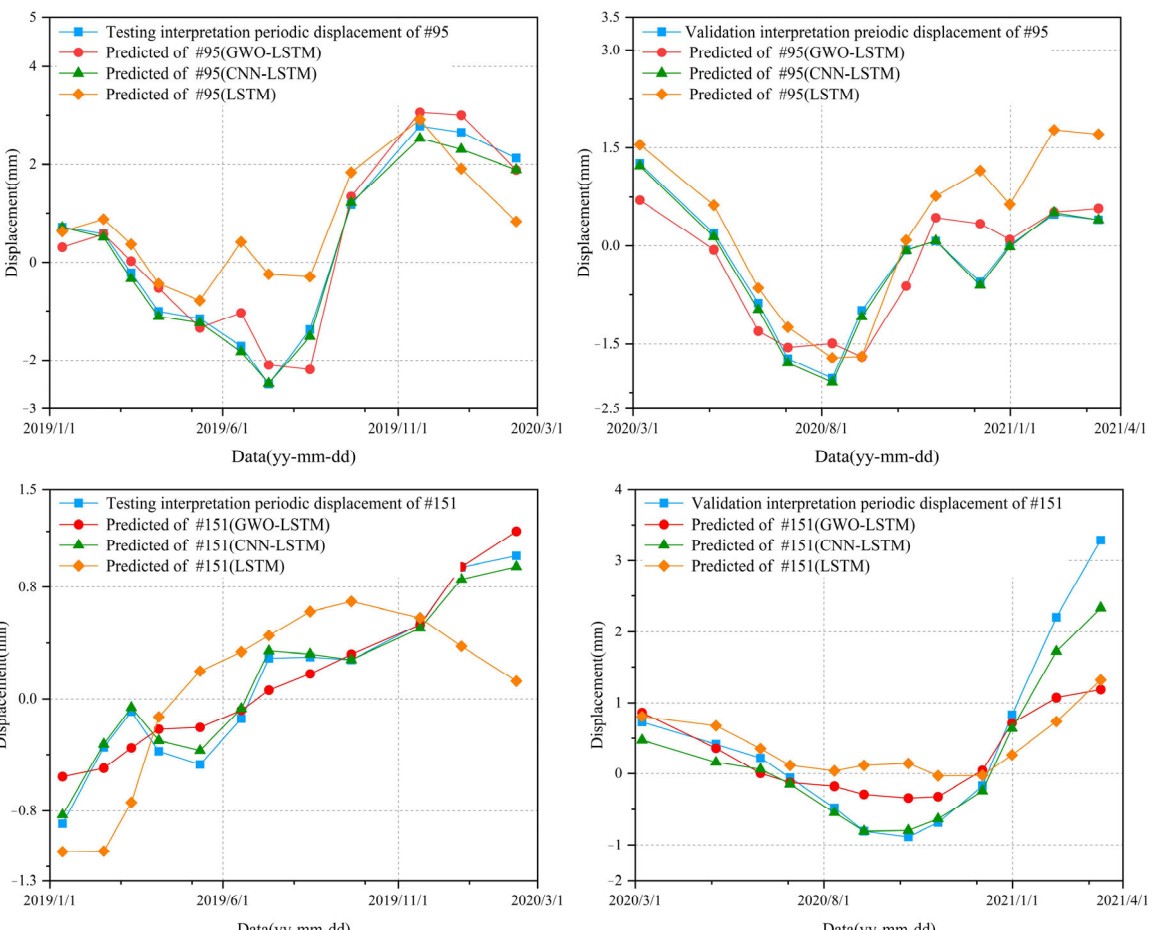

**Figure 14.** Prediction results of the periodic displacement.

**Table 3.** The periodic displacement prediction accuracy of towers #95 and #151.

| Tower Number | Model | Testing RMSE | Testing $R^2$ | Validation RMSE | Validation $R^2$ |
|---|---|---|---|---|---|
| #95 | GWO–LSTM | 0.413 | 0.941 | 0.470 | 0.734 |
| | CNN–LSTM | 0.161 | 0.991 | 0.056 | 0.996 |
| | LSTM | 1.092 | 0.586 | 0.834 | 0.161 |
| #151 | GWO–LSTM | 0.205 | 0.863 | 0.213 | 0.560 |
| | CNN–LSTM | 0.061 | 0.988 | 0.338 | 0.923 |
| | LSTM | 0.516 | 0.135 | 0.871 | 0.480 |

Figure 15 displays the results of the predictions made for the random items, and the RMSE and $R^2$ of the prediction model for random displacement are shown in Table 4. It can be seen that CNN–LSTM performs better than GWO–LSTM and LSTM models, in which the $R^2$ in both the training and validation sets is more significant than 0.9. The LSTM model performs poorly in multi-factor prediction, and the prediction results are less referential.

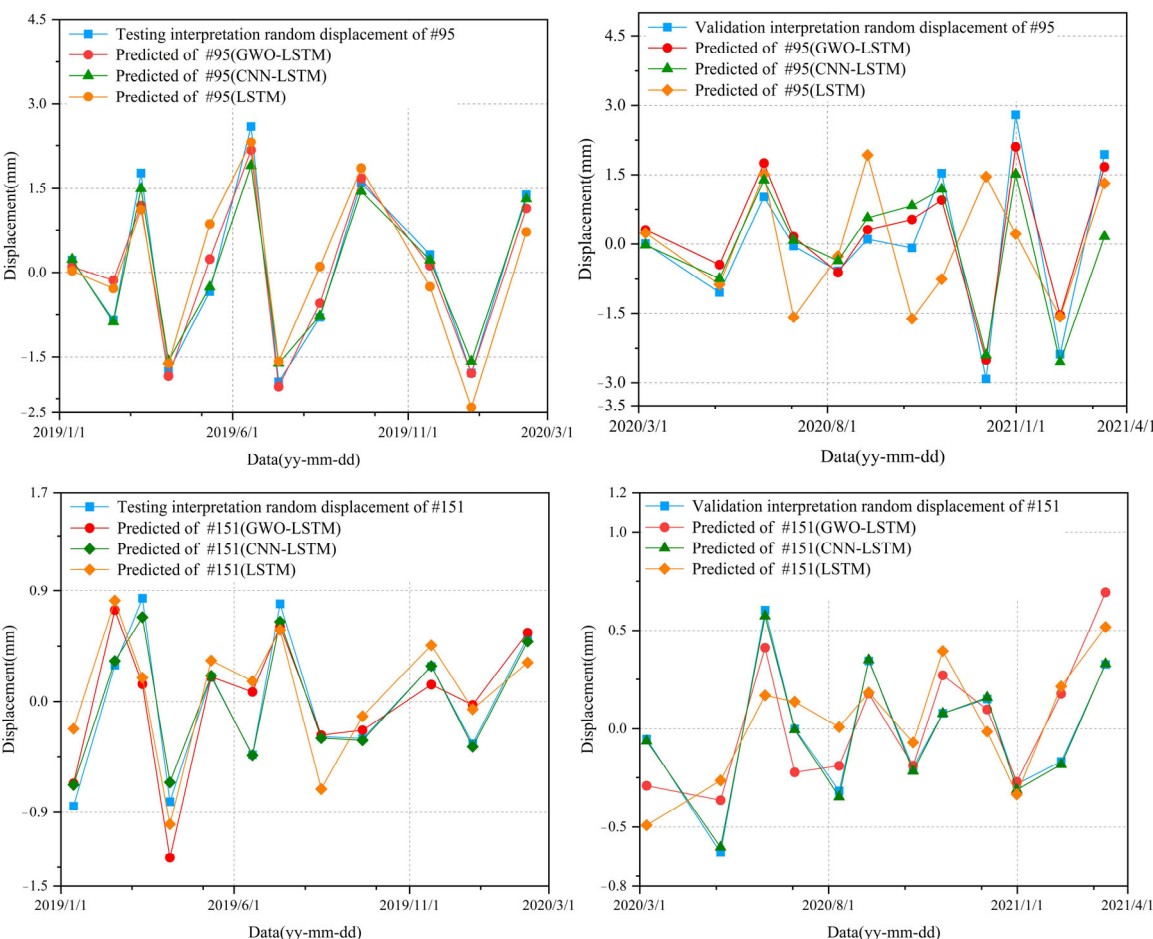

**Figure 15.** Prediction results of the random displacement.

**Table 4.** The random displacement prediction accuracy of towers #95 and #151.

| Tower Number | Model | Testing RMSE | Testing $R^2$ | Validation RMSE | Validation $R^2$ |
|---|---|---|---|---|---|
| | GWO–LSTM | 0.360 | 0.940 | 0.518 | 0.895 |
| #95 | CNN–LSTM | 0.259 | 0.969 | 0.251 | 0.971 |
| | LSTM | 0.612 | 0.828 | 0.739 | 0.786 |
| | GWO–LSTM | 0.336 | 0.631 | 0.214 | 0.564 |
| #151 | CNN–LSTM | 0.018 | 0.997 | 0.092 | 0.973 |
| | LSTM | 0.399 | 0.481 | 0.288 | 0.213 |

The CNN–LSTM model performs best, suggesting that it has a greater generalization capacity for multi-factor prediction based on periodic and random prediction results.

### 4.7. Cumulative Displacement Prediction and Accuracy Assessment

The transmission tower cumulative displacement was determined by cumulating the values of the periodic, trend, and random displacements. (Figure 16). The results show that the cumulative deformation of the transmission tower optimized by the GWO and CNN is consistent with the remote sensing interpretation results (Table 5). In #95, the RMSE and $R^2$ of the GWO–LSTM model test set are 0.93 mm and 0.986, respectively. The RMSE and $R^2$ values of the validation set were 1.412 and 0.923, respectively. For #151, the RMSE and $R^2$ of the CNN–LSTM model test set were 0.723 mm and 0.976, respectively, and the RMSE and $R^2$ of the validation set were 0.485 mm and 0.972, respectively. In addition, in the #95 prediction results, the displacement fluctuation of the transmission tower caused

by external disturbances is also reflected in the results of the GWO–LSTM and CNN–LSTM models. This indicates that the optimized LSTM can better predict the actual deformation of the transmission tower.

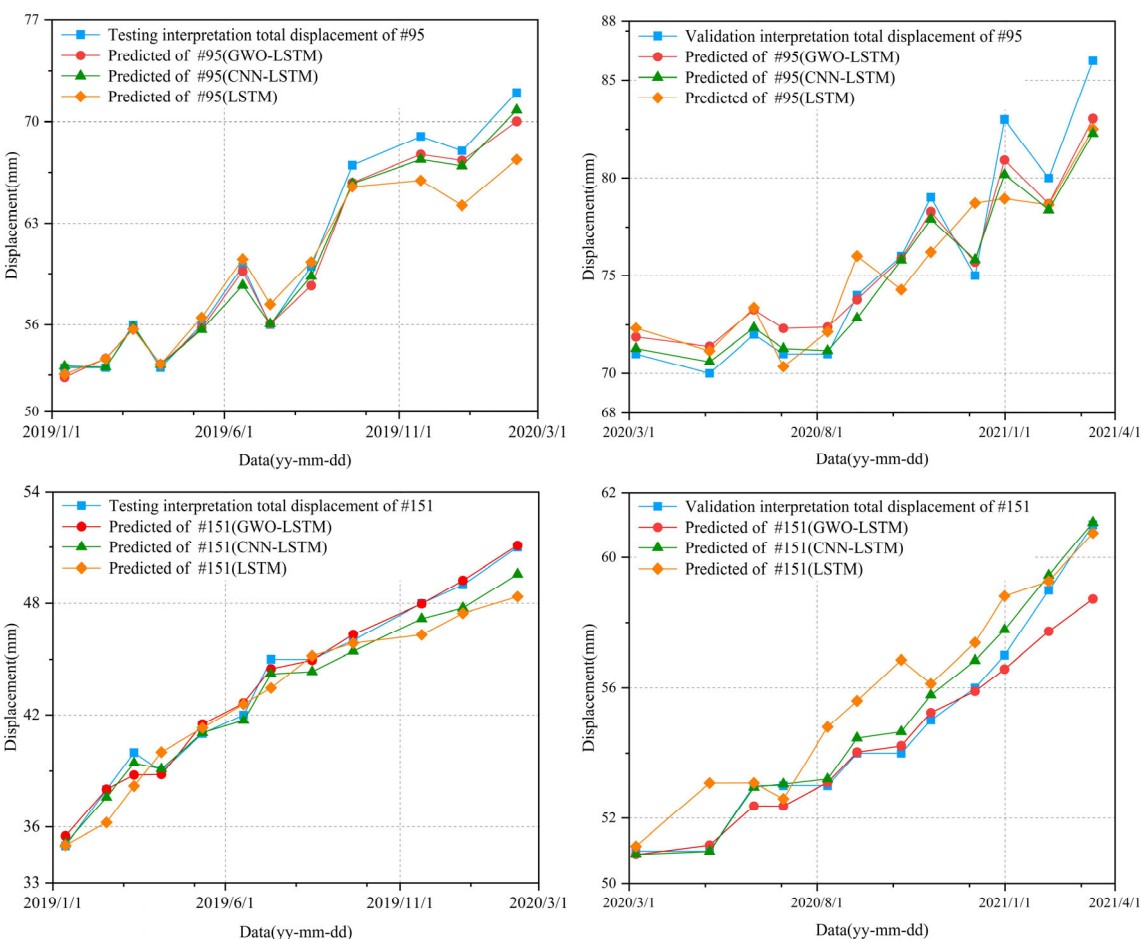

**Figure 16.** Prediction results of the total displacement.

**Table 5.** The total displacement prediction accuracy of towers #95 and #151.

| Tower Number | Model | Testing RMSE | Testing $R^2$ | Validation RMSE | Validation $R^2$ |
|---|---|---|---|---|---|
| #95 | GWO–LSTM | 0.930 | 0.986 | 1.412 | 0.923 |
|  | CNN–LSTM | 0.875 | 0.983 | 1.564 | 0.810 |
|  | LSTM | 2.054 | 0.905 | 1.568 | 0.809 |
| #151 | GWO–LSTM | 0.490 | 0.989 | 0.813 | 0.922 |
|  | CNN–LSTM | 0.723 | 0.976 | 0.485 | 0.972 |
|  | LSTM | 1.370 | 0.913 | 1.446 | 0.755 |

## 5. Discussion

Owing to fragile geological and environmental conditions, the transmission tower located in the Qarhan Salt Lake area of the Qinghai–Tibet Plateau will continue to be affected by extreme climates in the future. However, a lack of medium- and long-term monitoring means early warning research for transmission line towers in the study area is needed. This study attempts to use the MT-InSAR method to monitor deformation and obtain excellent time series deformation results of the ground subsidence of transmission towers. A deep learning model and a system of ground subsidence trigger factors were constructed. At the same time, early warning of transmission tower ground subsidence in

the study area was carried out based on the time series ground subsidence deformation data of representative towers.

### 5.1. Application of the MT-InSAR Technology

As shown in Figure 1, the MT-InSAR method and the division of time series results for different deformation states were the basis for the forecasting framework. The results of the remote sensing interpretation show that the ground subsidence of transmission lines is mainly concentrated in the central Salt Lake area. Moreover, there was an apparent subsidence funnel near the transmission towers. K-shaped clustering was used to reveal the ground subsidence deformation characteristics of the transmission tower under different geological and environmental conditions [51]. From Figure 7, it can be observed that the K-shape algorithm can obtain good clustering results for the time series ground subsidence deformation of the transmission tower. Combined with the field survey results, the overall upward type (Figure 7b) is mainly located in the desert, which might be affected by wind–sand accumulation. The horizontal fluctuation type (Figure 7c) was mainly located in the wilderness area, which might have been caused by external disturbances. The middle Salt Lake region, which was the primary deformation area, is where the majority of the overall downward type (Figure 7d) is spread.

### 5.2. Triggering Factors of the Study Area

To accurately forecast ground deformation, it is essential to choose appropriate triggering conditions [28,63]. Quantification is challenging because of the microscopic properties of saline soil and the influence of human engineering operations. Because tower foundation groundwater monitoring equipment is not installed, there is a lack of useful groundwater data. Therefore, in this study, more practical and reasonable rainfall and temperature time series data were used as prediction trigger factors for the transmission towers. Because the salt expansion effect in the study area is relatively small, the temperature change causes the fluctuation of groundwater and changes the essential stress conditions [20,43,46]. According to the MIC value shown in Figure 12, it can be noticed that changes in temperature and rainfall have a greater impact on # 95, which may also be related to # 95 being in the salt pond. The data source used in this study was Sentinel-1A, which can better meet the requirements of surface deformation monitoring accuracy [64]. However, owing to the triggering of the atmosphere and monitoring period, the correlation between more comprehensive trigger factors and ground deformation still needs to consider multi-source monitoring equipment [65]. At the same time, more external trigger factors should be explored, such as vehicle load, wind load, and mining frequency of mineral salt in the study area.

### 5.3. Ground Subsidence Prediction of the Transmission Towers

Currently, there is no literature on ground subsidence prediction research on particular geological conditions and transmission towers in the Salt Lake area. This is a new attempt and has achieved good results. Because it can preserve and use past data and fully utilize its advantages of extracting correlation information, the LSTM model offers an ideal prediction capacity for displacement time series [32]. However, the built-in parameter space of the LSTM model renders the prediction results highly uncertain. In this study, two optimization models (GWO and CNN) were used to determine the best hyperparameter combination to increase the precision and generalizability of the prediction outcomes [40,54]. Based on the two selected representative transmission towers from Table 3 to Table 5, it can be found that the optimized LSTM model has better prediction results. In future studies, the performance of the prediction models will need to be further studied under different environmental conditions, data support, and control factors.

## 6. Conclusions

This study attempts to use the MT-InSAR method to monitor deformation and obtain excellent time series deformation results of the ground subsidence of transmission towers. K-shape clustering was used to cluster the time series ground subsidence deformation trend of more than 300 base towers along the line. Combined with field investigation and representative deformation characteristics, # 95 and # 151 were selected for deep learning displacement prediction. The conclusions are summarized below:

(i) The mid-Salt Lake region is where the ground subsidence in the study area is most concentrated, according to MT-InSAR data. The time series results of the transmission towers exhibit three apparent features according to the K-shape clustering results. The negative effect of the deformation curve of the overall downward type on the transmission towers was the largest.

(ii) The MIC values show that #95 in the salt pond was more significantly affected by temperature and rainfall. This indicated that the towers in the salt pond were more susceptible to external factors and deformation.

(iii) The results of the ground subsidence prediction show that the LSTM optimized by CNN and GWO performs well in displacement prediction. The GWO–LSTM model was more suitable for trend prediction, whereas CNN–LSTM performed better under multiple factors.

Limited by the site conditions and fundamental data of the study area, this study has not yet made a more detailed experimental analysis of the deformation factors of saline soil in the study area. Nevertheless, the research presented in this study could provide a new idea for installing future monitoring equipment and deformation monitoring in the Salt Lake area. It could also be used in the monitoring and early warning of geological disasters in transmission networks.

**Author Contributions:** Writing and original draft preparation: B.J.; Writing—review and editing: K.Y. and L.G.; Methodology: T.Z.; Investigation: T.Y., B.G., B.Z. and Q.L. All authors have read and agreed to the published version of the manuscript.

**Funding:** Funded by Science and technology project of State Grid Corporation of China (52280721000A) "Research and application of large deformation mechanism and prevention technology of tower foundation in Salt Lake area", Contract No. SGQHDKYOSBJS2100034.

**Data Availability Statement:** The datasets are unavailable due to privacy and ethical restrictions.

**Acknowledgments:** We wish to thank the State Grid Qinghai Electric Power Research Institute and the Research Institute of Transmission and Transformation Projects, China Electric Power Research Institute, State Grid Corporation of China, Beijing.

**Conflicts of Interest:** The authors declare no conflict of interest.

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
