# Peer review of "The Prediction of Transmission Towers’ Foundation Ground Subsidence in the Salt Lake Area Based on Multi-Temporal Interferometric Synthetic Aperture Radar and Deep Learning"

_remotesensing, doi:10.3390/rs15194805_

Round 1
Reviewer 1 Report
1. The abstract needs to be rewritten. The issue is that it only briefly analyzed the outline of the article and did not describe the article’s prominent work, such as the relevant results, key contributions, and any notable findings.
2. The introduction needs a rearrangement of content as it lacks logical coherence. For example, the first paragraph indicates that early warning research in the Salt Lake area is still limited, but there is no further review or discussion of this problem. In the second paragraph, it begins to describe another issue, which appears to result in poor logical coherence.
3. Line 44-46 is too brief and requires a more detailed explanation.
4. The third section “Methodology” needs to be rewritten. For instance, regarding the MT-InSAR sub-section, it should be clarified that SBAS-InSAR is a specific implementation of MT-InSAR used for specific areas. MT-InSAR encompasses various multi-temporal interferometric SAR methods, and the choice of used method typically depends on the monitoring task requirements and data availability. This paper did not clearly distinguish the two different concepts. Similarly, concerning K-means and K-shape, they are two distinct clustering methods employed for different types of data. K-means is suitable for general multidimensional data, whereas K-shape is specifically designed for processing time series data to identify shape-similar patterns. Despite the similarity in their names, they have significantly different principles and application domains.
5. The issues mentioned in the above comment (comment 4) should be addressed throughout the entire paper.
6. The heading of section 4.1 cannot be simply “MT-InSAR results”.
7. The paper lacks sufficient details in certain areas. It does not thoroughly explore the damage and impact of saline soil on the transmission lines within the research area. The complex environmental conditions in the research area present specific challenges and difficulties, but these aspects are not elaborated upon, which could serve as an introduction to further discussions, including the introduction of scientific questions. However, relevant results are not detailed. In the methods section, the connection between these methods and other content is overlooked, and the data is briefly mentioned in just one sentence. Please refer to the following relevant papers for addressing these issues, and consider consulting other relevant publications.
Wang, R., Wang, X., Liu, H., Wang, Y., Peng, Y., Sun, W., & Liu, J. (2018, November). Rockfall hazard identification and assessment of the Langxian-Milin section of the transmission line passage of Central Tibet Grid Interconnection Project. In IOP Conference Series: Earth and Environmental Science (Vol. 189, No. 5, p. 052046). IOP Publishing.
Lan, H., Tian, N., Li, L., Wu, Y., Macciotta, R., & Clague, J. J. (2022). Kinematic-based landslide risk management for the Sichuan-Tibet Grid Interconnection Project (STGIP) in China. Engineering Geology, 308, 106823.
Wang, Y.; Cui, X.; Che, Y.; Li, P.; Jiang, Y.; Peng, X. Identification and Analysis of Unstable Slope and Seasonal Frozen Soil Area along the Litang Section of the Sichuan–Tibet Railway, China. Remote Sens. 2023, 15, 1317. https://doi.org/10.3390/rs15051317
Author Response
Dear Reviewer 1,
We sincerely appreciate your careful reading and valuable comments to improve this manuscript. In view of your rigorous review of the article and the problems in the article, we have carried out serious thinking and solved all the problems raised by the reviewers. The responses to the reviewers ' opinions are as follows.
In addition, we have made some revisions to the manuscript, including combing the overall thinking structure of manuscript, correcting format errors, modifying English expressions, and correcting grammatical errors. All revisions are highlighted in red font.
Main concern:
Concern 1: The abstract needs to be rewritten. The issue is that it only briefly analyzed the outline of the article and did not describe the article’s prominent work, such as the relevant results, key contributions, and any notable findings.
Response: Thank you for your insightful suggestion. In the revised abstract, we have elaborated on our principal findings and contributions and highlighted any findings that are particularly noteworthy. This will facilitate a clearer and more comprehensive understanding for the readers about the core content and significance of our paper. We have incorporated these modifications in the appropriate sections of the manuscript, specifically on lines 29-33, for your reference. We are confident that these revisions have enriched the abstract, making it more informative and reflective of the key focal points and contributions of the paper.
Concern 2: The introduction needs a rearrangement of content as it lacks logical coherence. For example, the first paragraph indicates that early warning research in the Salt Lake area is still limited, but there is no further review or discussion of this problem. In the second paragraph, it begins to describe another issue, which appears to result in poor logical coherence.
Response: Thanks for your comment. In response to your feedback, we have reorganized the content in the introduction to improve the logical flow. Specifically, we have expanded on the discussion related to the limitations of early warning research in the Salt Lake area, providing a more thorough review of this problem and bridging the gap to the subsequent content. We have also worked on ensuring a smoother transition between paragraphs to address the abrupt shift in topics, which you highlighted. We have marked the revised and supplemented sections in red in the Introduction section for your convenience in identifying the changes made.
Concern 3: Line 44-46 is too brief and requires a more detailed explanation.
Response: Thanks for your suggestion. We have made a supplementary in the appropriate position of the paper, please see line 50-54.
Concern 4: The third section “Methodology” needs to be rewritten. For instance, regarding the MT-InSAR sub-section, it should be clarified that SBAS-InSAR is a specific implementation of MT-InSAR used for specific areas. MT-InSAR encompasses various multi-temporal interferometric SAR methods, and the choice of used method typically depends on the monitoring task requirements and data availability. This paper did not clearly distinguish the two different concepts. Similarly, concerning K-means and K-shape, they are two distinct clustering methods employed for different types of data. K-means is suitable for general multidimensional data, whereas K-shape is specifically designed for processing time series data to identify shape-similar patterns. Despite the similarity in their names, they have significantly different principles and application domains.
Response: Thank you for your thoughtful feedback. As the commenter notes, the MT-INSAR and SBAS-INSAR have significantly different principles and application domains. Therefore, in order to effectively distinguish between the two concepts, the paper provides additional clarifications in lines 56-59 and 195-197. At the same time, the title of section 3.1 was changed to “SBAS-InSAR method”.
As stated by the reviewer, the K-means is suitable for general multidimensional data, whereas K-shape is specifically designed to work with time-series data to recognise patterns of similar shape. From the time-series deformation results of ground settlement of transmission line towers in Fig. 7a, it can be found that the time-series deformation shows different patterns of change. Therefore, as an effective time series clustering analysis method, the K-form algorithm can effectively obtain the deformation law of ground settlement of transmission line towers in the study area, and can distinguish the influence of different time series deformation results on the stability of transmission towers. In order to avoid the confusion between K-mean algorithm and K-form algorithm, this paper is supplemented in lines 214-216.
Concern 5: The issues mentioned in the above comment (comment 4) should be addressed throughout the entire paper.
Response: Thanks for your comment. We've combed through the full text and changed it accordingly.
Concern 6: The heading of section 4.1 cannot be simply “MT-InSAR results”.
Response: Thanks for your suggestion. After considering this section together, the title was changed to "Transmission tower ground deformation results", please refer to the section 4.1.
Concern 7: The paper lacks sufficient details in certain areas. It does not thoroughly explore the damage and impact of saline soil on the transmission lines within the research area. The complex environmental conditions in the research area present specific challenges and difficulties, but these aspects are not elaborated upon, which could serve as an introduction to further discussions, including the introduction of scientific questions. However, relevant results are not detailed. In the methods section, the connection between these methods and other content is overlooked, and the data is briefly mentioned in just one sentence. Please refer to the following relevant papers for addressing these issues, and consider consulting other relevant publications.
Response: Thank you for your recommendation. Image feature extraction and recognition technology is mature and widely used in landslide recognition. The extraction of disaster distribution by image data can greatly reduce the cost of field work, and it is of great significance in the prevention and management of geological disasters. However, due to the different deformation mechanisms and characteristics between land subsidence and landslide, there is few applications in land subsidence image recognition at present. How to identify land subsidence by the texture characteristics is still a big challenge. At present, we are interested in studying the identification of the hidden points of land subsidence from remote sensing images. The literatures provided by reviewer provide us with valuable ideas. Thank reviewer again for the literature provided, please consult references [1-2], and [12].
[1] Lan, H., Tian, N., Li, L., Wu, Y., Macciotta, R., & Clague, J. J. (2022). Kinematic-based landslide risk management for the Sichuan-Tibet Grid Interconnection Project (STGIP) in China. Engineering Geology, 308, 106823.
[2] Wang, R., Wang, X., Liu, H., Wang, Y., Peng, Y., Sun, W., & Liu, J. (2018, November). Rockfall hazard identification and assessment of the Langxian-Milin section of the transmission line passage of Central Tibet Grid Interconnection Project. In IOP Conference Series: Earth and Environmental Science (Vol. 189, No. 5, p. 052046). IOP Publishing.
[12] Wang, Y.; Cui, X.; Che, Y.; Li, P.; Jiang, Y.; Peng, X. Identification and Analysis of Unstable Slope and Seasonal Frozen Soil Area along the Litang Section of the Sichuan–Tibet Railway, China. Remote Sens. 2023, 15, 1317. https://doi.org/10.3390/rs15051317

Reviewer 2 Report
Dear authors, thank you for the manuscript. It concerns displacement prediction on power towers in an unstable area.
The manuscript is generally comprehensible and well written, I only have minor comments:
- section 3.1: why do you use SBAS method? PS method would probably give better results for power towers as these are good reflector. SBAS methods use spatial filtering, which is not desirable. Also, PS method does not need phase unwrapping which is potentially errorneous: unwrapping in time is done intrinsically and unwrapping in space is not needed when investigating spatially separated points.
- formula (2) applies only if the displacement happens truly in the vertical direction
- line 191: please specify clearly 67*2 images. Are these 67 dates, and 2 images/frames for each date?
- section 3.2: 67 images/dates is not many. Please discuss the reliability/accuracy of the method
- section 3.3: please explain the abbreviations
- section 4.1: please comment on the reference point so that a reader can be sure that some towers are really subject to uplift. Please comment also InSAR algorithm/implementation used
- line 294: not clear what it means "convert to positive values" and why
- section 4.5: please describe in more detail the formation of training, testing and validation sets, are they really separated? how many points were used for training and how many for validation? did you use shorter time series (of various length) for validation, based on the comparison with later measurements?
The language is generally comprehensible and easy to read, I also have only minor comments:
- line 215: missing stop after [29]
- line 314: difference between twelve days
- line 323: displacement that during
- line 435: is more obvious for temperature
Author Response
Response to Reviewer 2 Comments
Dear Reviewer 2,
We sincerely appreciate your careful reading and valuable comments to improve this manuscript. In view of your rigorous review of the article and the problems in the article, we have carried out serious thinking and solved all the problems raised by the reviewers. The responses to the reviewers ' opinions are as follows.
In addition, we have made some revisions to the manuscript, including combing the overall thinking structure of manuscript, correcting format errors, modifying English expressions, and correcting grammatical errors. All revisions are highlighted in red font.
Main concern:
Concern 1: section 3.1: why do you use SBAS method? PS method would probably give better results for power towers as these are good reflector. SBAS methods use spatial filtering, which is not desirable. Also, PS method does not need phase unwrapping which is potentially errorneous: unwrapping in time is done intrinsically and unwrapping in space is not needed when investigating spatially separated points.
Response: Thank you for your insightful comment regarding the use of the PS-InSAR method over the SBAS method for analyzing power towers. We acknowledge the advantages of the PS method in dealing with good reflectors and its inherent ability to handle phase unwrapping, thus potentially reducing errors associated with this process. However, after careful consideration, we decided to employ the SBAS-InSAR method in our study for several interconnected reasons. Firstly, the PS-InSAR method, while advantageous in certain scenarios, demands a substantial amount of sentinel data and is ideally suited for areas with excellent interferometric conditions, such as urban environments. Our study area, on the other hand, presents varying degrees of vegetation, potentially impacting the interferometric effect, and posing challenges for the PS-InSAR method.
Additionally, the SBAS-InSAR method allows for optimal utilization of available interferometric image pairs and exhibits lower data requirements on the time series, proving beneficial especially when dealing with an extensive range of data time phases. This flexibility and adaptability of the SBAS-InSAR method often yield better results under such circumstances. Moreover, prior research in the Salt Lake area has demonstrated the efficacy of the SBAS-InSAR method for ground subsidence monitoring, establishing a precedent for its application in this region [1-3]. Adopting the same methodological approach enables consistency and comparability with previous findings, which we believe enhances the value of our study. Based on these considerations and the specific characteristics of the Salt Lake area, we concluded that the SBAS-InSAR method offers a more stable and reliable monitoring effect for our research objectives. We appreciate the opportunity to clarify our methodological choice and remain open to further discussions and insights to improve our work.
[1] Xiang, W.; Liu, G.; Zhang, R.; Pirasteh, S.; Wang, X.; Mao, W.; Li, S.; Xie, L. Modeling Saline Mudflat and Aquifer Deformation Synthesizing Environmental and Hydrogeological Factors Using Time-Series InSAR. Ieee J. Sel. Top. Appl. Earth Observ. Remote Sens. 2021, 14, 11134-47, doi:10.1109/JSTARS.2021.3123514
[2] Xiang, W.; Zhang, R.; Liu, G.; Wang, X.; Mao, W.; Zhang, B.; Cai, J.; Bao, J.; Fu, Y. Extraction and analysis of saline soil deformation in the Qarhan Salt Lake region (in Qinghai, China) by the sentinel SBAS-InSAR technique. J. Geod. Geodyn. 2021, doi:10.1016/j.geog.2020.11.003
[3] Xiang, W.; Zhang, R.; Liu, G.; Wang, X.; Mao, W.; Zhang, B.; Fu, Y.; Wu, T. Saline-Soil Deformation Extraction Based on an Improved Time-Series InSAR Approach. Isprs Int. J. Geo-Inf. 2021, 10, 112, doi:10.3390/ijgi10030112
Concern 2: formula (2) applies only if the displacement happens truly in the vertical direction.
Response: Thanks for your comment. In the context of our study, most of the transmission towers, as depicted in Fig. 1(a), are situated in the hinterland of the Qaidam Basin. This area is characterized by its flat terrain and minimal height differences, which significantly reduces the likelihood of geological disasters involving slope-directed motion deformation such as landslides. Furthermore, comprehensive site surveys of the transmission towers and assessments of the deformation patterns of surrounding features corroborate our assumption. These field observations indicate that the ground subsidence in the study area primarily manifests in the vertical direction, as illustrated in Fig. 1(b). Given these specific geographic and geological characteristics of the study area, we deemed the application of Equation (2) for radar line-of-sight projection to be a reasoned and justified approach in this instance. We appreciate your scrutiny and will ensure to clarify this aspect in the revised manuscript to avoid any potential misinterpretation.
Concern 3: line 191: please specify clearly 67*2 images. Are these 67 dates, and 2 images/frames for each date?
Response: We thank for you pointing this error out. For a more accurate presentation, we have made additions “For the SBAS-InSAR calculation, 67*2 (2 images for each date) Sentinel-1A images were obtained.”, line 208-209.
Concern 4: section 3.2: 67 images/dates is not many. Please discuss the reliability/accuracy of the method
Response: Thank you for bringing up the concern regarding the number of images/dates used in our study and its potential impact on the reliability and accuracy of our method. While it is generally true that a greater number of data points can enhance the learning efficacy of a time series prediction model, we are also bound by constraints such as the historical deformation of transmission towers in the study area and the availability of data. In this context, we managed to acquire 67 effective SAR images through freely accessible open-source channels, which, based on previous studies [1, 2], have been demonstrated to yield satisfactory prediction results. Moreover, to address the challenges associated with a smaller dataset, we employed advanced deep learning models, namely CNN-LSTM and GWO-LSTM, for our modelling prediction. These models are particularly adept at learning from limited samples, thereby mitigating the risks of reduced data availability. The generalization ability of our chosen models was rigorously tested through predictive analysis at two different monitoring points. The high prediction accuracy achieved by the final model, with a Validation R^2 value of 0.972, attests to its reliability and robustness, indicating that the dataset used in this study is indeed capable of producing valid and reliable predictions.
[1] Yu, C.; Huo, J.; Li, C.; Zhang, Y. Landslide Displacement Prediction Based on a Two-Stage Combined Deep Learning Model under Small Sample Condition. Remote Sens. 2022, 14, 3732, doi:10.3390/rs14153732
[2] Wen, H.; Xiao, J.; Xiang, X.; Wang, X.; Zhang, W. Singular spectrum analysis-based hybrid PSO-GSA-SVR model for predicting displacement of step-like landslides: a case of Jiuxianping landslide. Acta Geotech. 2023, doi:10.1007/s11440-023-02050-9
Concern 5: section 3.3: please explain the abbreviations
Response: Thanks for your comment. We have added the appropriate comments to the abbreviations, and the changes are highlighted in red font.
Concern 6: section 4.1: please comment on the reference point so that a reader can be sure that some towers are really subject to uplift. Please comment also InSAR algorithm/implementation used.
Response: Thanks for your comment. In order to further demonstrate the significance of the sample towers for the study, they have been described accordingly in the paper, see the red-marked section in section 4.1.
Concern 7: line 294: not clear what it means "convert to positive values" and why.
Response: Thanks for your comment. In this study, the deformation of transmission towers in the focal area is primarily influenced by ground settlement. For analyzing ground deformation, we project the satellite line-of-sight direction to obtain the results in the vertical direction, as detailed on lines 203-206. It's important to note that in the MT-InSAR deformation results, negative values indicate ground subsidence. To facilitate subsequent modeling and prediction, and to maintain consistency and clarity in representation, we represent the settlement value of the tower base deformation as a positive value. To enhance the clarity and avoid any potential confusion, we have revised the phrasing in the manuscript to: "In order to visualise the ground subsidence prediction model, absolute valuation of the transmission tower ground subsidence displacement values was carried out", line 318-319.
Concern 8: section 4.5: please describe in more detail the formation of training, testing and validation sets, are they really separated? how many points were used for training and how many for validation? did you use shorter time series (of various length) for validation, based on the comparison with later measurements?
Response: Thanks for your comment. Thank you for your comments. We have explained the composition of the training, testing and validation sets sample data in the paper, please see the sections marked in red in Section 4.3 of the paper.
Concern 9: line 215: missing stop after [29].
Response: Thanks for your comment. We added that stop before [39].
Concern 10: line 314: difference between twelve days.
Response: We thank for you pointing this error out. We have changed it to “Combined with deformation data time span, the temperature difference between Sentinel-1A images two scenes was selected for research”, line 338-339.
Concern 11: line 323: displacement that during.
Response: We thank for you pointing this error out. We have changed it to “the displacement changes during the last image”, line 348.
Concern 12: line 435: is more obvious for temperature.
Response: We thank for you pointing this error out. We have changed it to “it can be noticed that changes in temperature and rainfall have a greater impact on # 95”, line 459-460.

Round 2
Reviewer 1 Report
N.A.